# Preoperative Predictors of Neoplasia in Patients Undergoing Small Bowel Resection for Complicated Crohn’s Disease: A Multicentre Case-Control Study

**DOI:** 10.3390/cancers15072004

**Published:** 2023-03-28

**Authors:** Capucine Chappe, Cecile Salut, Aurelien Amiot, Delphine Gaye, Nora Frulio, Bruno Lapuyade, Lucine Vuitton, Romain Altwegg, Cyrielle Gilletta, Mathurin Fumery, Guillaume Bouguen, Melanie Serrero, Maria Nachury, Nicolas de Suray, Ludovic Caillo, Mireille Simon, David Laharie, Pauline Rivière, Florian Poullenot

**Affiliations:** 1CHU de Bordeaux, Hôpital Haut-Lévêque, Service d’Hépato-Gastroentérologie et Oncologie Digestive, Université de Bordeaux, F-33000 Bordeaux, France; 2CHU de Bordeaux, Hôpital Haut-Lévêque, Service de Radiologie Digestive, Université de Bordeaux, F-33000 Bordeaux, France; 3Département des Maladies de l’Appareil Digestif, CHU Bicêtre, AP-HP, Université Paris Est Creteil, F-94000 Créteil, France; 4Department of Gastroenterology, University Hospital of Besançon, 3 Boulevard Fleming, F-25030 Besançon, France; 5Department of Gastroenterology, Saint-Eloi Hospital, F-34000 Montpellier, France; 6Department of Gastroenterology, CHU Toulouse, F-31300 Toulouse, France; 7Department of Gastroenterology, CHU Amiens, PeriTox UMR I-01, Université de Picardie Jules Verne, F-80000 Amiens, France; 8Department of Gastroenterology, CHU Rennes, F-35000 Rennes, France; 9Department of Gastroenterology, CHU Marseille, F-13005 Marseille, France; 10Department of Gastroenterology, CHU Lille, F-59000 Lille, France; 11Grand Hôpital de Charleroi, Site Saint-Joseph, Service d’Hépato-Gastroentérologie, B-6060 Charleroi, Belgium; 12Department of Gastroenterology, University Hospital of Nimes, F-30900 Nimes, France; 13Department of Gastroenterology, CH Pau, F-64000 Pau, France

**Keywords:** Crohn’s disease, small bowel neoplasia, predictive factors, small bowel adenocarcinoma

## Abstract

**Simple Summary:**

Crohn’s disease (CD) is associated with an increased risk of small bowel neoplasia (SBN). We aimed to assess preoperative predictors of SBN in CD patients. We conducted a retrospective case-control study including CD patients who underwent surgery: cases were diagnosed with SBN on histopathological analysis in 12 tertiary centres and controls had no neoplasia. Preoperative cross-sectional imaging was reviewed by a panel of blinded expert radiologists. Fifty cases were matched to one hundred and fifty consecutive controls. Old age, long small bowel CD duration, and stricture predicted the presence of SBN, particularly adenocarcinoma when patients have digestive wall thickening > 8 mm on preoperative imaging.

**Abstract:**

Crohn’s disease (CD) is associated with an increased risk of small bowel neoplasia (SBN). We aimed to assess preoperative predictors of SBN in CD patients. We conducted a retrospective case-control study including CD patients who underwent surgery: cases were diagnosed with SBN on histopathological analysis and controls had no neoplasia. Preoperative cross-sectional imaging was reviewed by a panel of blinded expert radiologists. Fifty cases were matched to one hundred and fifty consecutive controls. In multivariable analysis, predictors of SBN were age ≥ 50 years (OR = 28, 95% CI = 5.05–206), median CD duration ≥ 17.5 years (OR = 4.25, 95% CI = 1.33–14.3), and surgery for stricture (OR = 5.84, 95% CI = 1.27–35.4). The predictors of small bowel adenocarcinoma were age ≥ 50 years (OR = 5.14, 95% CI = 2.12–12.7), CD duration ≥ 15 years (OR = 5.65, 95% CI = 2.33–14.3), and digestive wall thickening > 8 mm (OR = 3.79, 95% CI = 1.45–11.3). A predictive score based on the aforementioned factors was constructed. Almost 73.7% of patients with a high score had SBA. Old age, long small bowel CD duration, and stricture predicted the presence of SBN, particularly adenocarcinoma when patients have digestive wall thickening > 8 mm on preoperative imaging.

## 1. Introduction

Crohn’s disease (CD) is an inflammatory bowel disease (IBD) characterized by an inappropriate inflammatory response in the gut [1]. In the context of chronic intestinal inflammation, CD is associated with an increased risk of colorectal cancer [2] and, to a lesser extent, small bowel neoplasia (SBN) [3], particularly in the ileum [4]. Small bowel adenocarcinoma (SBA) is the most common SBN in the general population and accounts for 5% of digestive tract cancers. It has a poor prognosis, with a 5-year overall survival rate of 14–33% [5]. Other SBNs include high-grade dysplasia, neuroendocrine tumours, gastrointestinal stromal tumour and lymphoma.

A recent population-based cohort study from Denmark and Sweden [6] found a nine-fold higher risk of SBN in CD patients (approximately one case in three hundred and eighty-five) than in the general population. The risk was higher in patients with paediatric-onset disease, ileal stricture, and a recent diagnosis. The risk of death due to SBN was seven-fold higher in CD patients than the general population. The CESAME cohort [7] showed a standardized relative risk of SBA of 34 for CD patients with small bowel involvement compared to the general population, and 46 for CD patients with disease duration longer than 8 years. In the recent multicentre NADEGE cohort [4] of 347 patients with SBA, the prevalence of CD was 8.7%. Patients with CD and SBA were younger than those with SBA but without CD. Beyond SBA, CD patients have an increased risk of lymphoma when taking thiopurine [8,9] and possibly anti-tumour necrosis factor agents [10]. Moreover, Epstein–Barr virus (EBV) is significantly associated with lymphoma development in IBD patients [11], particularly primary intestinal lymphoma, which accounts for 22–75% of lymphoma cases in IBD patients [12,13].

A preoperative diagnosis of SBN in CD patients is challenging. Indeed, a prospective cohort study showed that endoscopic screening for SBN was ineffective in patients at high risk of SBA (>10 years without bowel resection) [14]. Moreover, SBN is difficult to diagnose on cross-sectional imaging, and is usually diagnosed intra- or post-operatively [15,16,17]. Conversely, the therapeutic management of CD and SBN is significantly affected by the presence of malignancy.

Because of the increased risk of cancer in young CD patients, the lack of an effective screening strategy, and the poor prognosis associated with tumours, we aimed to determine preoperative predictors of SBN in patients with complicated CD.

## 2. Materials and Methods

### 2.1. Study Design and Patients

We conducted a retrospective, multicentre, case-control observational study in 11 French and 1 Belgium tertiary IBD centre.

The cases included adults with small bowel CD, diagnosed according to standard criteria [18], who underwent small bowel or ileocolonic resection for a stricture and/or penetrating complication between 2006 and 2021, and were diagnosed with SBN based on histopathological examination of the surgical specimen. The SBNs included SBA, high-grade dysplasia, neuroendocrine tumour, gastrointestinal tumour (GIST), and lymphoma. The cases were identified from the prospective databases of study centres.

The controls were adults diagnosed with small bowel CD based on the standard criteria, who underwent small bowel or ileocolonic resection for a stricture and/or penetrating complication between 2008 and 2021 with no evidence of SBN on histopathological examination of the surgical specimen. The controls were selected by searching the local pathological database of Bordeaux University Hospital for consecutive patients who underwent small bowel resection between 2008 and 2021.

We excluded patients for whom data from abdominal cross-sectional imaging performed in the year before the surgery were not available, as well as those who underwent surgery for refractory inflammatory CD.

### 2.2. Data Collection

The inclusion date corresponded to the surgery date. The medical records of patients were retrospectively reviewed, and the age at IBD diagnosis, sex, smoking status, phenotype according to the Montreal classification [19], indications for surgery (stricture or penetration), extra-intestinal manifestations, previous treatment (e.g., amino-salicylates, steroids, thiopurines, methotrexate, anti-tumour necrosis factor, vedolizumab, or ustekinumab), and number of previous bowel resections and perianal surgeries were extracted.

We also recorded the SBN characteristics, including the histological type (adenocarcinoma, lymphoma, neuroendocrine tumour, GIST, or high-grade dysplasia), tumour site (ileum, jejunum, or duodenum), cancer stage according to the TNM system [20] (where possible), and microsatellite instability status.

The results of the final laboratory tests performed before the inclusion were also recorded, including C-reactive protein, ferritin, haemoglobin, carcinoembryonic antigen, and lactate dehydrogenase levels. We also recorded surgical data, including surgical technique (laparotomy or laparoscopy), length of resection (cm), number of lymph nodes analysed, and the type of anastomosis performed.

The management of patients after the tumour diagnosis, need for surgical revision, and mortality (related or unrelated to the SBN) were recorded.

### 2.3. Radiological Assessments

An image scoring sheet was developed for analysis of the radiological images. After several preliminary discussions based on a systematic literature review of the CT and MRI criteria for CD and SBN, as well as current recommendations and radiologists’ opinions, we devised a list of 11 important imaging features [21,22]. The items were selected based on the consensus view of four radiologists (BL, CS, DG, and NF) with more than 10 years of experience in CT and MRI interpretation in digestive radiology, and three gastroenterologists (FP, CC, and PR). The items included stricture (yes or no), stricture appearance (regular or irregular), maximum parietal wall thickness (mm), sub-mucosal parietal oedema (yes or no), presence of nodes > 1 cm (yes or no), tissue infiltration (yes or no), infiltration of mesenteric fat (yes or no), abscess or fistula (yes or no), length of inflammatory involvement (cm), and upstream dilatation > 28 mm (yes or no). Several sequences of the MRI scans were analysed (including T1, T2, and post- gadolinium contrast) and the characteristics of parietal wall enhancement were recorded (low, moderate, or intense; homogeneous or stratified). Furthermore, if two cross-sectional imaging examinations were performed, we analysed whether there was a significant increase in node size or in the thickening wall, a modification of the infiltration of the mesenteric fat, a modification of the contrast uptake, or an upstream dilatation. Based on the aforementioned findings, the radiologists rated suspicion of SBN on a 10-point (1–10) scale.

The final preoperative cross-sectional imaging examination (CT or MRI) performed in the year before inclusion in the study was analysed for the case and control groups. All imaging were interpreted by a radiologist blinded to the clinical information.

### 2.4. Outcomes

The primary objective of the study was to determine the preoperative predictors of SBN in patients with complicated CD. The secondary objectives were to determine the preoperative predictors of SBA in patients with complicated CD and develop a preoperative predictive score for SBA in patients with complicated CD.

### 2.5. Statistical Analysis

Descriptive analysis was performed. Categorical variables are presented as frequencies (%) and were compared using the Chi-square test or Fisher’s exact test, depending on the expected numbers in each category. Continuous variables are presented as medians with interquartile ranges (IQRs) and were compared between cases and controls using the Wilcoxon rank sum test (significance level (α) = 0.05).

To validate the method used to analyse the images, we randomly selected 10 imaging scans that were analysed by all 4 radiologists. We evaluated interobserver agreement based on the intraclass correlation coefficient, calculated using the one-factor method (variable individuals and fixed raters), and yielding a single rating. The median Kappa value was 0.63, which is adequate for this type of study.

To identify predictors of SBN and SBA, we performed logistic regression analysis using backward selection, and used the likelihood ratio test for analysing cases and controls with SBN or SBA as the event of interest. Factors with *p*-values < 0.20 in the univariable analysis were included in the multivariable analysis. Continuous variables were categorized as quartiles, and adjacent categories were combined when the SBN rates were similar.

We constructed a predictive scoring system that assigns points to each item according to the value of the coefficient estimates in the multivariable logistic regression model. We determined cut-off values to classify patients according to their risk of SBA (low, intermediate, or high risk).

### 2.6. Ethical Concerns

This study was conducted in accordance with the principles of good clinical practice and the Declaration of Helsinki. As required by national recommendations, the study protocol was approved by the CNIL (Commission Nationale de l’Information et des Libertés—the French Data Protection Agency; no.: 4444632) and our Institutional Review Board.

## 3. Results

### 3.1. Patient Characteristics

In total, 50 cases with SBN on the histopathological evaluation of the surgical specimen were recruited from the 12 study centres. The control group included 150 consecutive patients from the University Hospital of Bordeaux. The characteristics of the study participants are presented in Table 1. The CD location, age at diagnosis, and treatment regimen did not differ significantly between the groups. The study participants included 95 (48%) females and 76 (39%) active smokers. The median body mass index of the study participants was 20.7 (18.5–23.3) kg/m^2^.

Patients with SBN were more likely to have a family history of IBD (8 (17%) vs. 10 (6.7%); *p* = 0.042), extra-intestinal manifestations (13 (27%) vs. 21 (14%); *p* = 0.036), and strictures (31 (62%) vs. 55 (37%); *p* < 0.001) compared to controls. Patients with SBN were older at the inclusion date (median age, 50.5 (41.0–59.8) vs. 34.0 (26.2–42.8) years; *p* < 0.001) and had a longer median IBD duration (19.7 (6.2–27.8) vs. 7.5 (2.5–13) years; *p* < 0.001) compared to controls.

### 3.2. Surgical Procedure

The surgical details of the study participants are presented in Table 2. Surgery was performed due to a symptomatic stricture in 68% (34/50) and 48% (72/150) of SBN cases and controls, respectively (*p* = 0.014). The SBN group had a lower albumin level at the inclusion date (32.2 (27.4–35.0) vs. 34.0 (31.0–38.0) g/L; *p* = 0.003), more frequent use of laparoscopy (39/50 (80%) vs. 92/150 (62%); *p* = 0.022), and a higher number of analysed nodes (17.5 (6.0–21.2) vs. 2.0 (1.0–6.0); *p* < 0.001) compared to controls.

### 3.3. Cross-Sectional Imaging

Table 3 presents the preoperative cross-sectional imaging features. SBN patients had greater parietal wall thickness (11.0 (9.0–13.0) vs. 10.0 (7.0–12.0) mm; *p* = 0.013), more frequent tissue infiltration (17/50 (34%) vs. 28/150 (19%); *p* = 0.025), and more frequent stricture irregularity (19/50 (39%) vs. 28/150 (19%); *p* = 0.006) compared to controls.

When a MRI had been performed, parietal enhancement at the latest time was more intense in the control group (in controls than cases (25/57 (44%) vs. 2/8 (20%), respectively; *p* = 0.014). The median score for suspicion of neoplasia was 6 (3.0–8.0) and 2 (0.0–3.0) in the SBN and control groups, respectively (*p* < 0.001). The radiologists suspected cancer (i.e., score > 5/10) in 26 (52%) SBN patients.

### 3.4. SBN Characteristics

Table 4 presents the SBN characteristics of the participants. SBN was diagnosed on histopathological examination of the surgical specimen in 49% of the cases (29/50), and during intra-operative examination in 33% of the cases (16/50) (Figure 1). The remaining cases were diagnosed during endoscopy (5/50; 10%) or based on highly suspicious imaging characteristics (3/50; 6%). SBA was the most common histopathological diagnosis (36; 73%) (Figure 2). At the inclusion date, most tumours were T3 or T4 stage (35; 83.3%); 17 (38.5%) demonstrated lymph node involvement (N1–3), and 12 (27%) demonstrated metastasis. Positive surgical margins (R1–2) were excised in 22% of patients. Postoperative adjuvant chemotherapy was administered to 30 (65%) patients. The median follow-up duration was 133 (28.6–297.1) weeks. In January 2022, 16/50 (32%) patients had died and the median age of death was 50 (41.0–61.2) years.

### 3.5. Risk Factors of SBN

The multivariable Cox regression analysis showed that age ≥ 50 years at the inclusion date (odds ratio [OR] = 28, 95% confidence interval [CI] = 5.05–206; *p* < 0.001), surgery performed due to stricture (OR = 5.84, 95% CI = 1.27–35.4; *p* = 0.034), and median CD duration ≥ 17.5 years at the inclusion date (OR = 4.25, 95% CI = 1.33–14.3; *p* = 0.015) were independently associated with SBN at time of surgery (Table 5). Conversely, the presence of mesenteric fat infiltration on CT (OR = 0.27, 95% CI = 0.07–0.93; *p* = 0.043) was independently associated with a lower risk of SBN.

### 3.6. SBA Characteristics and Risk Factors for SBA

Of the 36 SBA patients, 12 (33%) demonstrated signet ring cells and 15 (47%) demonstrated well-differentiated cells. The disease stage was T3–4 in 35 (89%) cases, lymph nodes were involved in 17 (38.6%) cases, and metastases were present in 12 (31%) cases. Positive surgical margins (R1–2) were excised in 26% of patients. Details regarding the SBA are presented in Table 6. In January 2022, 16/36 (44%) patients had died (median age of 50 (41.0–61.2) years). The multivariable Cox regression analysis showed that age ≥ 50 years at the inclusion date (OR = 5.14; 95% CI = 2.12–12.7; *p* < 0.001), median CD duration ≥ 15 years at inclusion (OR = 5.65, 95% CI = 2.33–14.3; *p* < 0.001), and a maximum parietal wall thickness > 8 mm (OR = 3.79, 95% CI = 1.45–11.3; *p* = 0.006) were independently associated with SBA at the time of surgery (Table 7) (Figure 3).

### 3.7. Development of a Score for Predicting the Presence of SBA

Based on the association between inflammation and SBA, we constructed a scoring system to predict the presence of SBA at the time of surgery. The predictive scoring system included three risk factors independently associated with the risk of adenocarcinoma. Each item was weighted based on the coefficient estimates of the multiple logistic regression model: age ≥ 50 years: 1 point; median CD duration ≥ 15 years: 2 points; and maximum parietal wall thickness > 8 mm: 1 point.

SBA was identified during surgery in 73.7% of patients with a score of 4, 36.4% of patients with a score of 3, and 9.1% of patients with a score of 0–2 (Figure 4).

## 4. Discussion

In this multicentre case-control study, we analysed preoperative predictors of SBN in patients with complicated CD. Age ≥ 50 years, surgery for stricture, CD duration ≥ 17.5 years, and the absence of mesenteric fat infiltration were independently associated with the presence of SBN. The most common tumour was SBA. The subgroup analysis showed that age ≥ 50 years, CD duration ≥ 15 years, and maximal parietal wall thickness > 8 mm were independently associated with a risk of SBA at surgery. The aforementioned clinical and imaging factors were used to develop a prediction score for SBA.

Limited evidence exists regarding the clinical presentation, diagnosis, and risk factors of SBN in IBD patients because of its rarity. Several risk factors have been suggested for SBN in CD [23], including male sex [15], CD duration [15], occupational exposure [15], surgery [24,25], and exposure to IBD treatment [15,26]. The results of our study are consistent with a recent study that found the highest risk of SBN in ileal [4,6,27] and stricturing CD [6]. Moreover, CD duration was a major risk factor for SBN in the present and previous studies [4,7,15,28].

In our study, the SBN patients were older and had a longer median time between CD diagnosis and surgery compared to controls. We found similar results for the adenocarcinoma sub-group, which was the most frequent SBN in our study. The development of adenocarcinoma follows the inflammation–dysplasia–cancer sequence, as suggested by the presence of dysplastic lesions around the cancer in resected specimens [29,30]. A previous study found that the pathogenesis of SBA is closely linked to IBD activity and duration [28,31], suggesting the need for caution in patients with complicated CD, long disease duration, or age ≥ 50 years. As previously described [6], few SBN patients were simultaneously diagnosed with CD, suggesting the presence of undiagnosed asymptomatic CD.

In the present study, neoplasia was not associated with exposure to immunomodulators; however, only 5/50 patients had lymphoma (10%).

Most patients (>80%) were diagnosed with SBN intraoperatively, or on the basis of histological analysis. In the remaining cases, the diagnosis was made based on endoscopy (10%) or highly suspicious imaging features (6%). Thus, cross-sectional imaging often fails to detect SBN as usually observed in the literature, and that it is therefore both clinical and morphological arguments that will guide the clinician [7,14,15,17,32,33].

One of the main strengths of our study was the central reading of morphologic examinations. Only a single retrospective study has analysed the morphological examinations of patients (*n* = 14) performed before the occurrence of SBA [34]. In that study, SBA in patients with CD was associated with bowel thickness > 10 mm, similar to our results. In fact, maximum parietal wall thickness > 8 mm was independently associated with the risk of adenocarcinoma. Moreover, we found that an absence of mesenteric fat infiltration was independently associated with SBN. These factors can be easily determined by radiologists in clinical practice and guide treatment strategies.

In our study, most patients (89%) were diagnosed at an advanced stage of disease (T3–4) and surgical margins were positive for malignancy in 26% of cases, similar to previous studies [4].

Moreover, 32% of patients with SBN (16/50) had died by January 2022 (all patients had SBA). In the present study, patients with SBA were younger at both the time of enrolment and the time of death (median age: 50 years) compared to previous studies [27]. These findings suggest that, in this context of poor prognosis in young patients, appropriate management by oncological surgery is essential. The completeness of resection and number of positive lymph nodes are important prognostic factors in SBA patients [35,36]; however, identifying high-risk patients is a major issue. Predictors of SBA were used to develop an easy-to-use clinical and imaging predictive scoring system. This system dichotomized the cohort into groups unlikely to develop SBA (9.1% of patients with a score < 2) and at high risk of SBA (73.7% of patients with a score of 4). After validation in a prospective cohort, our scoring system may aid the early diagnosis of SBA and determination of the need for oncologic surgery with lymph node removal.

Our study had several limitations. First, it was a retrospective chart review of a rare condition. Second, we enrolled cases from 11 centres that may have been heterogeneous in terms of clinical practice; this should be considered when interpreting our results. Third, we chose to study all of the small bowel neoplasia and not only adenocarcinoma. This can be justified by the fact that we wanted an answer to a practical clinical question: the question was whether patients would have a neoplastic lesion on the surgical specimen, regardless of the histological result. Our study also had several strengths. To our knowledge, this is one of the largest multicentre studies of the predictors of SBN in CD patients. Moreover, expert radiologists blinded to the clinical data analysed the scans using a standardized form. Another strong point is the development of an easy-to-use in clinical practice predictive score of SBA.

## 5. Conclusions

In summary, the present case-control study identified simple predictors of SBN and SBA in CD patients, namely old age, long disease duration, stricturing complications (for SBN only), and wall thickening (for SBA only). An easy-to-use predictive score was established to evaluate the risk of SBA, and help physicians and surgeons decide regarding oncological surgery. Further studies are needed to validate the identified predictors.

## Figures and Tables

**Figure 1 cancers-15-02004-f001:**
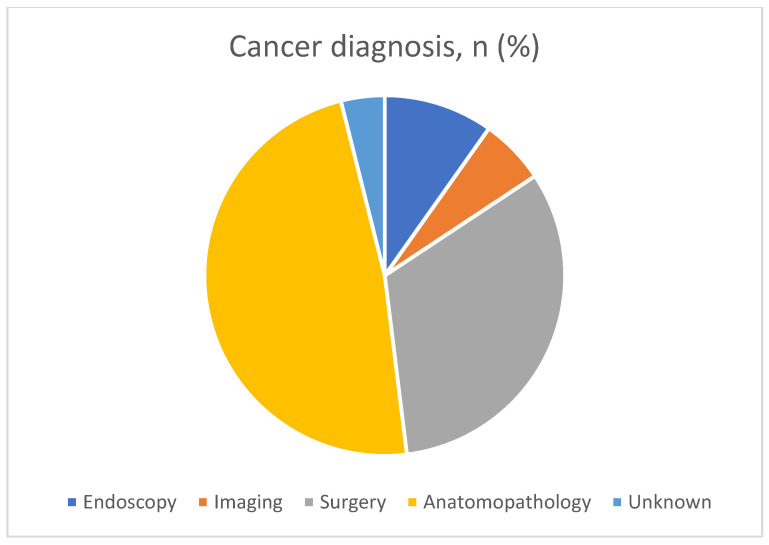
Cancer diagnosis in our study.

**Figure 2 cancers-15-02004-f002:**
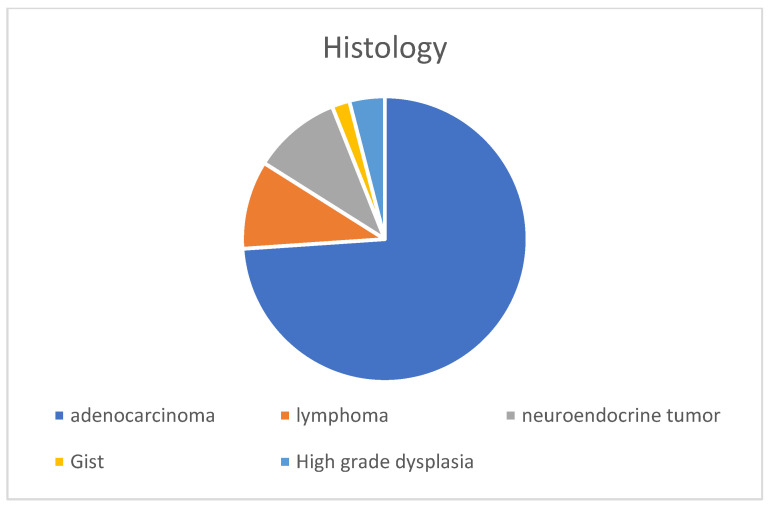
Histology of small bowel neoplasia.

**Figure 3 cancers-15-02004-f003:**
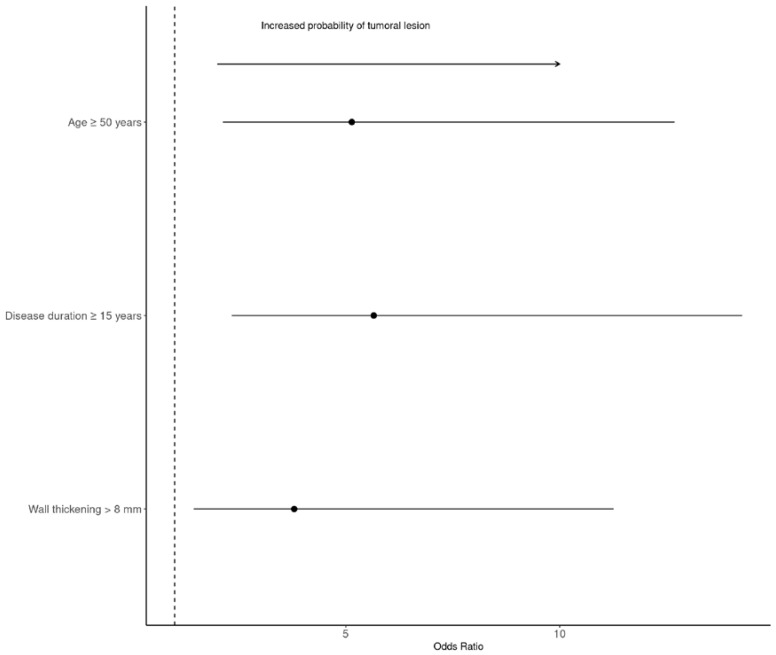
Forest plot of the probability of small bowel adenocarcinoma. Data are odds ratios; 95% confidence intervals are indicated by error bars.

**Figure 4 cancers-15-02004-f004:**
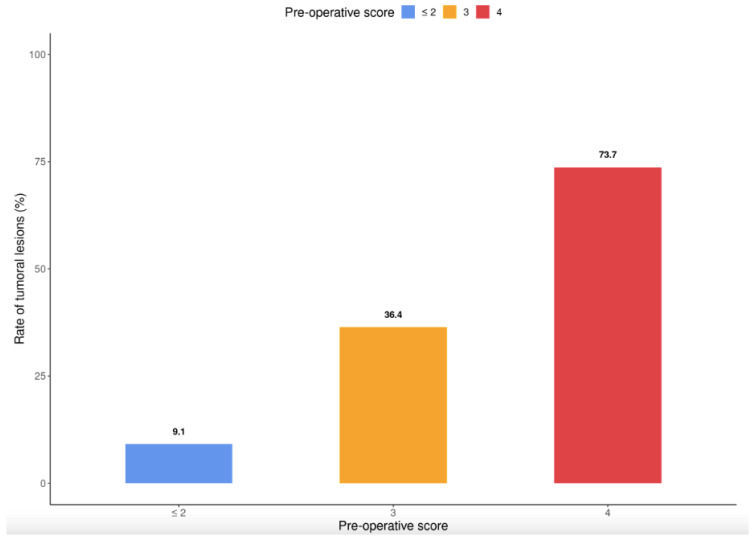
Scoring system for predicting the presence of SBA at the time of surgery.

**Table 1 cancers-15-02004-t001:** Demographic, clinical, and therapeutic characteristics of patients with and without SBN associated with CD.

	Overall, *n* = 200	Case, *n* = 50	Control, *n* = 150	*p*
Female sex, *n* (%)	95 (48%)	20 (40%)	75 (50%)	0.2
Median age, years (IQR)	38.0 (28.0–50.0)	50.5 (41.0–59.8)	34.0 (26.2–42.8)	<0.001
Body mass index, kg/m^2^ (IQR)	20.7 (18.5–23.3)	19.9 (18.5–22.8)	20.9 (18.5–23.6)	0.4
Smoker, *n* (%)	76 (39%)	13 (28%)	63 (43%)	0.068
Family history of IBD, *n* (%)	18 (9.1%)	8 (17%)	10 (6.7%)	0.042
Median CD duration, years (IQR)	8.5 (3.1–17.5)	19.7 (6.2–27.8)	7.5 (2.5–13.0)	<0.001
Age at diagnosis (years) °, *n* (%)				0.3
- A1 (<17)	18 (9.0%)	4 (8.2%)	14 (9.3%)	
- A2 (17–40)	160 (80%)	37 (76%)	123 (82%)	
- A3 (>40)	21 (11%)	8 (16%)	13 (8.7%)	
Disease location °, *n* (%)				0.4
- L1 (ileal disease)	133 (66%)	32 (64%)	101 (67%)	
- L2 (colonic disease)	1 (0.5%)	1 (2.0%)	0 (0%)	
- L3 (ileocolonic disease)	59 (30%)	15 (30%)	44 (29%)	
- L4 (isolated upper disease)	7 (3.5%)	2 (4.0%)	5 (3.3%)	
Behaviour °, *n* (%)				<0.001
- B1 (non-stricturing and non-penetrating)	5 (2.5%)	5 (10%)	0 (0%)	
- B2 (structuring)	86 (43%)	31 (62%)	55 (37%)	
- B3 (penetrating)	109 (55%)	14 (28%)	95 (63%)	
Perineal disease, *n* (%)	54 (27%)	12 (24%)	42 (28%)	0.6
Extra-intestinal manifestation, *n* (%)	34 (17%)	13 (27%)	21 (14%)	0.036
History of corticosteroid use, *n* (%)				0.090
Naive to corticoids, *n* (%)	44 (23%)	5 (10%)	39 (27%)	
Corticosensibility, *n* (%)	71 (36%)	23 (48%)	48 (33%)	
Corticosteroid dependence, *n* (%)	37 (19%)	8 (17%)	29 (20%)	
Corticosteroid resistance, *n* (%)	42 (22%)	12 (25%)	30 (20%)	
5ASA use, *n* (%)	16 (8.2%)	3 (6.5%)	13 (8.8%)	0.8
Immunosuppressant use, *n* (%)	119 (60%)	30 (61%)	89 (60%)	0.9
Thiopurines use, *n* (%)	37 (31%)	10 (33%)	27 (30%)	0.7
Methotrexate use, *n* (%)	12 (10%)	2 (7.1%)	10 (11%)	0.7
Adalimumab use, *n* (%)	22 (11%)	3 (6.2%)	19 (13%)	0.2
Infliximab use, *n* (%)	37 (19%)	8 (17%)	29 (19%)	0.7
Vedolizumab use, *n* (%)	2 (1.0%)	0 (0%)	2 (1.3%)	>0.9
Ustekinumab use, *n* (%)	6 (3.0%)	4 (8.3%)	2 (1.3%)	0.032
Previous surgery for IBD, *n* (%)	62 (31%)	14 (28%)	48 (32%)	0.6
Previous stricturoplasty, *n* (%)	5 (36%)	2 (22%)	3 (60%)	0.3

Data are median (interquartile range [IQR]) or frequency (%). IBD: inflammatory bowel disease; CD: Crohn’s disease. ° According to the Montreal classification.

**Table 2 cancers-15-02004-t002:** Details of surgical procedures in the two subgroups.

Surgical Characteristics	Overall, *n* = 200	Case, *n* = 50	Control, *n* = 150	*p*
Indication for surgery, *n* (%)				0.002
- fistula	38 (19%)	9 (18%)	29 (19%)	
- abscess	46 (23%)	4 (8.0%)	42 (28%)	
- stricture	103 (52%)	31 (62%)	72 (48%)	
- peritonitis	8 (4.0%)	2 (4.0%)	6 (4.0%)	
- others	5 (2.5%)	4 (8.0%)	1 (0.7%)	
Laboratory parameters at the time of surgery				
- haemoglobin, g/dL (IQR)	12.6 (11.4–13.6)	11.9 (11.0–13.4)	12.8 (11.6–13.7)	0.069
- ferritin, μg/L (IQR)	82.0 (33.0–160.0)	76.0 (25.5–166.2)	82.0 (36.5–160.0)	0.6
- albumin, g/L (IQR)	34.0 (31.0–38.0)	32.2 (27.4–35.0)	34.0 (31.0–38.0)	0.003
- CRP, mg/L (IQR)	17.0 (5.0–40.0)	25.0 (6.5–63.0)	14.0 (5.0–36.5)	0.2
Time between imaging and surgery, days (IQR)	32.0 (10.2–84.8)	44.0 (16.8–127.5)	29.5 (10.0–78.0)	0.090
Oncological surgery, *n* (%)	15 (7.5%)	13 (26%)	2 (1.3%)	<0.001
Length of resection, cm (IQR)	29.0 (20.0–40.0)	31.0 (20.0–45.0)	28.0 (19.0–38.0)	0.2
Stoma, *n* (%)	66 (33%)	22 (44%)	44 (29%)	0.056
Number of nodes analysed, *n* (%)	10.0 (2.00.0)	17.5 (6.0–21.2)	2.0 (1.0–6.0)	<0.001
Laparoscopy, *n* (%)	131 (66%)	39 (80%)	92 (62%)	0.022

Data are median (interquartile range [IQR]) or frequency (%).

**Table 3 cancers-15-02004-t003:** Radiological analysis of baseline images’ features.

	Overall, *n* = 200	Case, *n* = 50	Control, *n* = 150	*p*
Presence of a stricture, *n* (%)	189 (94%)	46 (92%)	143 (95%)	0.5
Maximum parietal wall thickness, mm (IQR)	10.0 (8.0–12.0)	11.0 (9.0–13.0)	10.0 (7.0–12.0)	0.013
Presence of parietal reinforcement, *n* (%)	184 (93%)	40 (83%)	144 (97%)	0.004
Presence of sub-mucosal parietal oedema, *n* (%)	125 (63%)	23 (46%)	102 (68%)	0.004
Presence of adenopathy > 1 cm, *n* (%)	55 (28%)	14 (28%)	41 (27%)	>0.9
Presence of mesenteric fat infiltration, *n* (%)	147 (74%)	29 (58%)	118 (79%)	0.004
Presence of tissue infiltration, *n* (%)	45 (22%)	17 (34%)	28 (19%)	0.025
Stricture irregularity, *n* (%)	47 (24%)	19 (39%)	28 (19%)	0.006
Multiple lesions, *n* (%)	8 (4%)	3 (6%)	5 (3,3%)	0.4
Abscess, *n* (%)	60 (30%)	13 (26%)	47 (32%)	0.5
Perforation, *n* (%)	9 (4.5%)	4 (8.0%)	5 (3.3%)	0.2
Fistula, *n* (%)	79 (40%)	20 (40%)	59 (39%)	>0.9
Length of inflammatory involvement, cm (IQR)	20.0 (11.0–30.0)	20.0 (11.5–35.0)	20.0 (11.0–30.0)	0.3
Upstream dilatation > 28 cm, *n* (%)	112 (56%)	33 (67%)	79 (53%)	0.072
If MRI had been performed				
Parietal enhancement on final MRI, *n* (%)				0.014
- low	2 (3.0%)	2 (20%)	0 (0%)	
- moderate	38 (57%)	6 (60%)	32 (56%)	
- intense	27 (40%)	2 (20%)	25 (44%)	
- stratified	38 (57%)	3 (30%)	35 (61%)	0.065
If two successive examinations were performed				
Significant increase in size of adenopathy, *n* (%)	2 (9.5%)	2 (22%)	0 (0%)	0.2
Increase in parietal thickening, *n* (%)	6 (30%)	3 (38%)	3 (25%)	0.6
Change in mesenteric fat infiltration, *n* (%)	9 (43%)	4 (44%)	5 (42%)	>0.9
Modification of contrast uptake, *n* (%)	3 (15%)	2 (25%)	1 (8.3%)	0.5
Upstream dilatation, *n* (%)	7 (33%)	3 (33%)	4 (33%)	>0.9
Score of 1–10 for radiological suspicion of cancer, *n* (IQR)	2.0 (1.0–3.2)	6.0 (3.0–8.0)	2.0 (0.0–3.0)	<0.001
Score > 5 for radiological suspicion of cancer, *n* (%)	31(16%)	26 (52%)	5 (3.3%)	<0.001

Data are median (interquartile range [IQR]) or frequency (%).

**Table 4 cancers-15-02004-t004:** Oncological details, management, and outcomes in the SBN group.

	*n* = 50
Cancer diagnosis, *n* (%)	
- Endoscopy	5 (10%)
- Imaging	3 (6.1%)
- Surgery	16 (33%)
- Anatomopathology	24 (49%)
- unknown	2 (4%)
Neoplasia location, *n* (%)	
- ileum	44 (90%)
- jejunum	3 (6.1%)
- duodenum	1 (2.0%)
- unknown	2 (4%)
Histology, *n* (%)	
- Adenocarcinoma	35 (71%)
- Adenocarcinoma + neuroendocrine tumour	1 (2.0%)
- Lymphoma	5 (10%)
- Neuroendocrine tumour	5 (10%)
- GIST	1 (2.0%)
- High-grade dysplasia	2 (4.1%)
Signet ring cells, *n* (%)	13 (27%)
Family history of cancer, *n* (%)	2 (4.5%)
Personal history of cancer, *n* (%)	2 (4.1%)
EBV, *n* (%)	0 (0%)
Differentiation of tumour, *n* (%)	
- well	21 (55%)
- moderate	14 (37%)
- low	3 (7.9%)
TNM: T, *n* (%)	
- T1–2	7 (16.7%)
- T3–4	35 (83.3%)
TNM: N, *n* (%)	
- N0	27 (61%)
- N1–2	15 (34%)
- N3	2 (4.5%)
Metastasis (>M1), *n* (%)	12 (27%)
Positive surgical margins [≥R1], *n* (%)	10 (22%)
Microsatellite instability status, *n* (%)	3 (8.6%)
Adjuvant chemotherapy, *n* (%)	30 (65%)
Palliative chemotherapy, *n* (%)	6 (13%)
Immunotherapy, *n* (%)	2 (4.3%)
0ncological surveillance, *n*(%)	14 (31%)
Death, *n* (%)	16 (32%)
Age at death, years (IQR)	50.0 (41.0–61.2)

Data are median (interquartile range [IQR]) or frequency (%).

**Table 5 cancers-15-02004-t005:** Predictors of small bowel neoplasia in multivariable and Cox regression analyses, *n* = 200.

Characteristic	OR	95% CI	*p*
Age ≥ 50 years	28.0	5.05–206	<0.001
Indication for surgery: stricture	5.84	1.27–35.4	0.034
Median Crohn’s disease duration ≥ 17.5 years	4.25	1.33–14.3	0.015
Presence of mesenteric fat infiltration	0.27	0.07–0.93	0.043

OR: odds ratio, CI: confidence interval.

**Table 6 cancers-15-02004-t006:** Oncological details, management, and outcomes in the adenocarcinoma subgroup.

Colonne1	*n* = 36
Diagnosis of neoplasia, *n* (%)	
- Endoscopy	3 (8.3%)
- Imaging	1 (2.8%)
- Surgery	14 (39%)
- Anatomopathology	17 (47%)
- unknown	1 (2.8%)
Neoplasia location, *n* (%)	
- ileum	33 (92%)
- jejunum	2 (5.6%)
- duodenum	1 (2.8%)
Signet ring cells, *n* (%)	12 (33%)
Family history of cancer, *n* (%)	2 (6.5%)
Personal history of cancer, *n* (%)	0 (0%)
Tumour differentiation, *n* (%)	
- well	15 (47%)
- moderate	14 (44%)
- low	3 (9.4%)
TNM: T, *n* (%)	
- T1–2	4 (11.1%)
- T3–4	32 (89%)
TNM: N, *n* (%)	
- N0	22 (61%)
- N1–2	12 (33%)
- N3	2 (5.6%)
Metastasis (>M1), *n* (%)	11 (31%)
Positive surgical margins [≥R1], *n* (%)	9 (26%)
Microsatellite instability status, *n* (%)	3 (10%)
Treatment, *n* (%)	
- Revision surgery	5 (14%)
- Adjuvant chemotherapy	22 (69%)
- Palliative chemotherapy	5 (16%)
- Immunotherapy, *n* (%)	1 (3.1%)
- Oncological surveillance, *n* (%)	8 (26%)
Death, *n* (%)	16 (44%)
Age at death, years (IQR)	50.0 (41.0–61.2)

Data are median (interquartile range [IQR]) or frequency (%).

**Table 7 cancers-15-02004-t007:** Predictors of SBA in multivariable and Cox regression analyses.

Characteristic	OR	95% CI	*p*
Age at the time of surgery ≥ 50 years	5.14	2.12, 12.7	<0.001
Median Crohn’s disease duration ≥ 15 years	5.65	2.33, 14.3	<0.001
Maximum parietal wall thickness > 8 mm	3.79	1.45, 11.3	0.006

OR: odds ratio, CI: confidence interval.

## Data Availability

The data underlying this article are available in the article.

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
