# Peer review of "Preoperative Predictors of Neoplasia in Patients Undergoing Small Bowel Resection for Complicated Crohn’s Disease: A Multicentre Case-Control Study"

_cancers, 2023, doi:10.3390/cancers15072004_

Round 1

Reviewer 1 Report

- Please replace the terms 'univariate' and 'multivariate' with 'univariable' and 'multivariable' respectively.

Author Response

March 24th, 2023

Dear Sir,

            Thank you for giving us the opportunity to submit a our version of our manuscript “Preoperative predictors of neoplasia in patients undergoing small bowel resection for complicated Crohn's disease: a multicentre case-control study” to be submitted for publication in Cancers.

We appreciate the observations and comments of the reviewers, which we believe enhance the overall quality of the manuscript. We have made the requested alterations in the text, and we provide here point-by-point answers to the specific questions.

Thank you in advance for considering the publication of our revised manuscript.

Yours sincerely,

Capucine Chappe. on behalf of the authors

Comments to the Authors:

 Please replace the terms 'univariate' and 'multivariate' with 'univariable' and 'multivariable' respectively.

Answer: Thank you for your comment. I have replace the item.

Reviewer 2 Report

It is a very interesting multicentere study about a significant topic. The suggested scoring system for predicting of small bowel neoplasia is very usefull and should be futher valiaded. 

Please find some suggestions: 
1. Please add information/data about the medications of patients with lymphoma and EBV status of these patients and analyze the data. Is there any association between use of anti-TNF and/or thiopourines?

2. Please add information/data about the activity of disease (CDAI or HB score). 

Author Response

March 24th, 2023

Dear Sir,

            Thank you for giving us the opportunity to submit a our version of our manuscript “Preoperative predictors of neoplasia in patients undergoing small bowel resection for complicated Crohn's disease: a multicentre case-control study” to be submitted for publication in Cancers.

We appreciate the observations and comments of the reviewers, which we believe enhance the overall quality of the manuscript. We have made the requested alterations in the text, and we provide here point-by-point answers to the specific questions.

Thank you in advance for considering the publication of our revised manuscript.

Yours sincerely,

Capucine Chappe. on behalf of the authors

Comments to the Authors:

1. Please add information/data about the medications of patients with lymphoma and EBV status of these patients and analyze the data. Is there any association between use of anti-TNF and/or thiopourines?

Thank you for your comment. 

- In the table 4: we didn't have any patient with EBV +. 

- Because of the lack of patients with lymphoma (just 5), we didn't realized statistical analyzed on this subgroup.

  • 1/5 patients with lymphoma was undergoing a combotherapy with thiopurines and infliximab at the moment of the diagnosis. 
  • 1/5 had a treatment with IFX but this treatment was stopped at the moment of the diagnostic of lymphoma.
  • 2/5 with lymphoma had a treatment with ADA but this treatment was stopped at the moment of the diagnostic of lymphoma.
  • 2/5 didn't have thiopurines, ada or ifx. 

2. Please add information/data about the activity of disease (CDAI or HB score). 

Indeed, it could have been interesting. Unfortunately we don't have information about the activity of disease (in relation to the retrospective nature of the study)

Reviewer 3 Report

Abstract:

We have here 2 sections. The first part, called "simple summary", is very well done and convincing for a potential reader. It seems to me a very good short visiting card of the manuscript.

The abstract itself is well structured, sufficiently concise, accompanied by 4 keywords that are well chosen and suggestive for the manuscript.

On a scale of 1 to 10, I’ll give 10 points for the abstract.

Introduction:

The introduction occupies less than one page, it is well done, but not impressive, it includes 16 bibliographic references; I did not find any deficiencies to report.

On a scale of 1 to 10, I agree 9 points for introduction.

Methodology:

This is a very good chapter, which contains all the necessary elements, each stage of the study is correctly and completely described, starting with the study design and ending with the statistical analysis of the data collected.

On a scale of 1 to 10, I agree 10 points for methodology.

Results:

The results are well presented, in the text, 7 tables and 2 figures. I followed the data carefully; I did not notice anything irregular. However, I have a suggestion, considering the abundance of numerical values ​​and percentages in the text, but especially from the 7 tables, I think that an at least partial graphic presentation of the results would benefit the potential reader.

On a scale of 1 to 10, I agree 8 points for results.

Discussion:

The discussion chapter is insufficient, in terms of the space allocated, just over one page, and in terms of the bibliographic references that support these discussions. The subject of the manuscript is sufficiently generously addressed in the literature to be able to easily supplement the bibliography, and the quality of the study carried out by the authors I believe requires a broader approach to the discussions.

In this situation, on a scale of 1 to 10, I agree 6 points for discussion.

Conclusion:

The conclusions are correctly written, on 5 lines of text, the authors briefly also indicate possible future research directions. Considering the quality of the study, I would have expected something more from the conclusions chapter, but, in its current form, it reaches an acceptable level.

On a scale of 1 to 10, I agree 8 points for conclusions.

Bibliography/References:

31 references, current, correctly written and correctly quoted in the text, I think they are still not enough for this manuscript, considering the suggestion above, regarding completing the discussions.

On a scale of 1 to 10, I agree 7 points for the bibliography.

Figures/Tables:

I identified 2 images and 7 tables, of good quality, with satisfactory resolution, which are necessary and useful for the manuscript. I think that some of the data presented in the tables also require graphical representations.

On a scale of 1 to 10, I agree 7 points for this chapter.

Review Decision:

Accept after minor revision.

Author Response

March 24th, 2023

Dear Sir,

            Thank you for giving us the opportunity to submit a our version of our manuscript “Preoperative predictors of neoplasia in patients undergoing small bowel resection for complicated Crohn's disease: a multicentre case-control study” to be submitted for publication in Cancers.

We appreciate the observations and comments of the reviewers, which we believe enhance the overall quality of the manuscript. We have made the requested alterations in the text, and we provide here point-by-point answers to the specific questions.

Thank you in advance for considering the publication of our revised manuscript.

Yours sincerely,

Capucine Chappe. on behalf of the authors

Comments to the Authors:

Reviewer: 3

Comments to the Author

Results:

The results are well presented, in the text, 7 tables and 2 figures. I followed the data carefully; I did not notice anything irregular. However, I have a suggestion, considering the abundance of numerical values ​​and percentages in the text, but especially from the 7 tables, I think that an at least partial graphic presentation of the results would benefit the potential reader.

Answer: Thank you for this comment. We’ve talked about it with our statistician and he thought it more appropriate to present the data in a table in order to be as comprehensive as possible.

Discussion:

The discussion chapter is insufficient, in terms of the space allocated, just over one page, and in terms of the bibliographic references that support these discussions. The subject of the manuscript is sufficiently generously addressed in the literature to be able to easily supplement the bibliography, and the quality of the study carried out by the authors I believe requires a broader approach to the discussions.

Answer:  Thank you for this comment. We modified part of the discussion and added bibliographic references.

31 references, current, correctly written and correctly quoted in the text, I think they are still not enough for this manuscript, considering the suggestion above, regarding completing the discussions.

Answer: Thank you for this suggestion. We added bibliographic references.

I identified 2 images and 7 tables, of good quality, with satisfactory resolution, which are necessary and useful for the manuscript. I think that some of the data presented in the tables also require graphical representations.

Answer: Thank you for this suggestion. We have added two graphical representations.
